# Virtual Screening-Based Study of Novel Anti-Cancer Drugs Targeting G-Quadruplex

**DOI:** 10.3390/pharmaceutics15051414

**Published:** 2023-05-05

**Authors:** Ruizhuo Ouyang, Jinyao Liu, Shen Wang, Weilun Zhang, Kai Feng, Conghao Liu, Baolin Liu, Yuqing Miao, Shuang Zhou

**Affiliations:** 1Institute of Bismuth and Rhenium Science, University of Shanghai for Science and Technology, Shanghai 200093, China; ouyangrz@usst.edu.cn (R.O.);; 2School of Health Science and Engineering, University of Shanghai for Science and Technology, Shanghai 200093, China; 3Cancer Institute, Tongji University School of Medicine, Shanghai 200092, China; shuangzhou@tongji.edu.cn

**Keywords:** G-quadruplexes, virtual drug screening, A549 cell, anti-cancer drugs, SHAFTS method

## Abstract

In order to develop new anti-cancer drugs more efficiently and reduce side effects based on active drug targets, the virtual drug screening was carried out through the target of G-quadruplexes and 23 hit compounds were, thus, screened out as potential anticancer drugs. Six classical G-quadruplex complexes were introduced as query molecules, and the three-dimensional similarity of molecules was calculated by shape feature similarity (SHAFTS) method so as to reduce the range of potential compounds. Afterwards, the molecular docking technology was utilized to perform the final screening followed by the exploration of the binding between each compound and four different structures of G-quadruplex. In order to verify the anticancer activity of the selected compounds, compounds 1, 6 and 7 were chosen to treat A549 cells in vitro, the lung cancer epithelial cells, for further exploring their anticancer activity. These three compounds were found to be of good characteristics in the treatment of cancer, which revealed the great application prospect of the virtual screening method in developing new drugs.

## 1. Introduction

The search for new and effective anti-cancer drugs was long pursued. Traditional drugs, such as Cisplatin and Adriamycin, usually target DNA by blocking its transcription and synthesis, thus causing apoptosis [1]. These drugs are not sufficiently target specific and inevitably have serious side effects [2]. In recent years, as more research on tumor signaling pathways emerged, many new anti-cancer drugs were designed using protein kinases or cell membrane receptors as specific targets in order to simultaneously reduce drug side effects and improve drug efficacy.

G-quadruplex is a high-level structure formed by folding DNA or RNA rich in tandem repeat guanine and the four guanine bases are connected by hydrogen bonds [3]. The structure of G-quadruplex depends on the orientation of the DNA strands and the syn and anti conformation of the guanines and can fold into a variety of different conformations according to environmental conditions [4], that varies by the position of adjacent loop regions. Depending on the 5′-3′ versus 3′-5′ orientation of segments in the self-folded oligonucleotide chain(s), G quadruplexes can be categorized as parallel structure, anti-parallel structure, or both [5]. With the development of chromatin immunoprecipitation and high-throughput sequencing technologies, G-quadruplex was proved to be present in telomeres and promoter regions of genes, and involved in important biological processes, such as telomere damage signaling, transcriptional regulation and mRNA translation and replication [6]. The high aggregation of G-quadruplex in promoter regions suggests its role in regulating gene expression. In addition, G-quadruplex is thought to have more relevance to cancer-related genes due to the overexpression of G-quadruplex in cancer cells. The structural heterogeneity of G-quadruplex and its high abundance in oncogene promoters make it an ideal target for anti-cancer drugs [6,7,8]. Telomere is a highly conserved G-rich repetitive DNA single strand at the end of eukaryotic chromosome, which can ensure the stability of chromatin. Normally, telomeres in somatic cells shorten during each cell cycle, eventually leading to apoptosis; in contrast, in most tumor cells, telomerase counteracts this effect, allowing tumor cell immortalization [9,10,11]. If appropriate small molecule can induce the 3′ ends of human telomeric DNA to fold up into G-quadruplex-ligand structures, the telomere will be unable to recognize the template and cannot add TTAGGG repeats to telomeres providing their elongation; subsequently, the cell will face age and death like somatic cells [9,12].

In recent years, an increasing number of compounds were found to stabilize G-quadruplexes such as Pyridostatin and CX-5461 for breast cancer [6]. The models of ligand interacting with G-quadruplex is tetrad-stacking, groove-binding and loop-binding [13]. Typical compounds are shown in Figure 1, where the compounds are commonly characteristic of a planar aromatic ring with the branched chains generally protonated and charged to form a stable stacked structure [14]. Although some of them entered clinical trials, there are still no relevant drugs on the market. The main reason for this situation is the lack of sufficient selectivity of the ligand for the G-quadruplex and the limitations in water solubility and stability.

To address these issues, virtual screening using computer-aided drug design was initiated to select drug candidates more efficiently and cost-effectively [15,16]. Virtual screening is one of the most critical technologies in computer-aided drug design, through with potentially effective candidate compounds can be selected from the database containing a large number of organic compounds, and the further experimental tests with these candidate compounds can avoid the blind testing of a large number of compounds by high-throughput screening, as a result of speeding up the development of new drugs. As we all know, much manpower and material resources are usually taken to develop a new drug. However, it is meaningful to save the initial research and reduce the development costs through virtual screening. As early as 1981, Fortune magazine believed that computer-aided drug design had good development prospects. Since the technology of virtual screening is constantly iterative and updated, the mainstream screening methods have successively formed including molecular docking, pharmacophore model, molecular similarity comparison as well as machine learning. In this work, three-dimensional (3D) molecular similarity calculation and molecular docking were mainly used.

It is a fundamental assumption about the ligand-based drug design that the ligands with similar two or three dimensional (2D or 3D) structures will have similar properties towards their targets. Herein, a method called shape feature similarity (SHAFTS) was used to carry out the 3D molecular similarity calculations [17,18], by introducing six template molecules as a basis for determining the degree of structural similarity, and performing an initial screening in the commercial database SPECS. The candidate compounds were submitted to the molecular docking simulations on the four major structurally characterized conformations of G-quadruplex for the further screening [19,20]. Finally, three of the identified compounds were selected for subsequent biological experiments to further validate their anti-cancer effects. By combining these methods, a set of ligands targeting G-quadruplex were identified with their anti-cancer activity validated, which lays the foundation for related studies.

## 2. Materials and Methods

### 2.1. Database Preparation

The SPECS compounds library was chosen as the molecular database for virtual screening [21], which included 208,332 compounds using Lipinski rule for the screening of compound databases to remove molecules not suitable for being potential drugs [22]. Compounds that conform to this rule usually have better pharmacokinetic properties and may also show better utilization in the metabolism process. As the molecular database was filtered by Lipinski rules, which means the compound in the databases already have good pharmacokinetic performance in theory, all compounds directly step into the screening process. All compounds containing inorganic atoms were removed and the remaining ones were added hydrogen, ionized at pH 5.1–pH 9.1. The conformational ensembles of the corresponding target molecules in the database were generated by means of Cyndi [23].

### 2.2. Query Molecules Selection

Based on the similarity principle, some ligands were selected with high affinity and selectivity toward DNA G-quadruplex, drug-like property, chemical diversity, different mechanism of interaction and in vitro activity in tumor cells as query molecules [14]. For example, D. Cian et al. surveyed the G-quadruplex ligands tested for telomerase inhibition [14,24]. S. Neidle critically examined the major classes of the currently developed quadruplex-binding small molecules [5,25,26]. The query molecules chosen (Figure 2) are the 3,6-bis(1-methyl-2-vinylpyridinium) carbazole diiodide63 (oBMVC:2) the pentacyclic acridine derivative (RHPS4 10), 128–130 (2-(4-(10H-indolo[3,2-b]quinolin-11-yl)piperazin-1-yl)-N,Ndimethylethanamine:19) 178 (CX-3543:5-fluoro-N-(2-((S)-1-methylpyrrolidin-2-yl)ethyl)-3-oxo-6-((R)-3-(pyrazin-2-yl), the (bisquinolinium) phenanthrolinecompound260 Phen-DC3 3,5-difluoro-4-hydroxybenzylidene imidazolinone a disubstituted benzofuranderivative.

### 2.3. 3D Similarity Calculation

Similarity calculations for molecules are available for the 2D MACCS method [27] and the 3D ROCS method [28]. The former uses a binary string form to represent molecular descriptors, where each bit in the value indicates whether (if the position is occupied) a particular feature is present in the corresponding molecule. The core idea of ROCS is to compare the similarity of molecular traits: the more similar two molecules are in shape, the larger the respective volume of the largest overlap that can be formed when they are folded together. SHAFTS is a recently introduced method that not only compares molecules in three dimensions, as in the ROCS method, but also performs the matching of pharmacophore groups.

Here, SHAFTS was used to calculate the 3D similarity between the molecular databases and the query. SHAFTS adopts a hybrid similarity metric of the shape-densities overlap (Shape Score) and chemical feature fit values (Feature Score) to score and rank alignment modes and target molecules. The shape score evaluates the degree of overlap between the spatial appearance of the molecule and the target molecule and the similarity of the chemical properties. In practice, a Gaussian function is used to describe the volume of each atom, colored according to its chemical identity, so that a corrected color-based matching term can be calculated alongside the overlapping volume, with the result normalized to [0, 1]. The features core is used to evaluate the degree of similarity between the target molecule and the pharmacophore extracted from the template and is also normalized to [0, 1]. At the same time, a certain weight is given to feature score and the combination of the two scores is the final similarity calculation.

The SHAFTS was used to search the conformation databases based on the query templates. The molecules with similarity greater than 1.0 were selected for further study. The similarity score and physical properties of these molecules were shown in Appendix A.

### 2.4. Molecular Docking

Molecular docking experiments were used to screen and design drugs by means of drug molecule–receptor interactions. The compounds selected in the previous step were submitted to ensemble docking simulations which were carried out by using AutoDock Vina. According to the DNA strand orientation, the structure of G-quadruplexes could be divided into four conformations: parallel, antiparallel and two mixed types with both parallel and antiparallel features including 143D, 1KF1, 2HY9, 2JPZ were used as receptors, among which 1KF1 (PDB Id of the G-quadruplexes) is a classical parallel structure, 143D an antiparallel structure and 2HY9 and 2JPZ, the mixed topological structures. The four structures were reported to be favorable to maximize the rationality and effectiveness of screening. All the candidate compounds were docked with the four G-quadruplexes one by one. AutoDock Vina uses a semi-flexible docking method. Binding of candidate ligands to G-quadruplexes targets in four conformations was evaluated on the basis of binding score, with smaller docking scores representing more stable binding. The compounds were ranked according to the scoring results and finally, 23 hit compounds were screened out. Afterwards, compounds 1, 6 and 7 of 23 hit compounds were selected for further verification experiments.

### 2.5. Cytotoxicity Assay

All reagents were purchased from Titan Technology Co., Ltd. (Shanghai, China) and Aladdin Reagents Ltd. (Shanghai, China). The cytotoxicity was evaluated via MTT assay. The A549 was seeded in sterile 96-well culture microplates at a density of 1 × 10^4^ cells/well, which were incubated with DMEM culture medium for 24 h at 37 °C with 5% CO_2_ until the flat bottom was covered with the cell monolayer. Then, 2.0 μL each of the compounds at different concentration was added into culture medium, and 5 wells were replicated at each concentration level. After the co-incubation for another 24 h, 20 μL of MTT solution (5 mg/mL, i.e., 0.5% MTT) was added to each well and continue incubating for 4 h. Then, the incubation was terminated with the culture solution carefully aspirated from the wells. A total of 100 μL of dimethyl sulfoxide (DMSO) was added to each well and the blue crystals were fully dissolved by shaking on the shaker for 10 min at low speed. The absorbance values of each well were measured at 570 nm on the ELISA. Blank groups (medium, MTT, DMSO) and control group (cells, same concentration of drug dissolution medium, culture medium, MTT, DMSO) were also set up. According to the absorbance, the cytotoxicity of cell in different time periods was analyzed.

### 2.6. In Vitro Inhibition of Compounds 1, 6 and 7

MTT assay was used for the inhibition determination of compounds 1, 6 and 7 towards cell proliferation in vitro. The initial density of A549 cells was 4 × 10^4^ cells/ml, which were seeded in sterile 96-well culture microplates. The cells were incubated with DMEM culture medium for 24 h at 37 °C with 5% CO_2_ until the cell monolayer was covered with flat bottom. Then, 2 μL each of compounds 1, 6 and 7 at different concentration gradient was added into culture medium, and 5–6 wells were replicated at each concentration. After the co-incubation for another 24 h, 20 μL of MTT solution (5 mg/mL, i.e., 0.5% MTT) was added to each well followed by a 4 h incubation. After the incubation was terminated and the culture solution was carefully aspirated from the wells, 100 μL DMSO was added to each well. The blue crystals were fully dissolved by shaking on the shaker for 10 min at low speed. The absorbance values of each well were measured at 570 nm. Blank groups (medium, MTT, DMSO) and control group (cells, same concentration of drug dissolution medium, culture medium, MTT, DMSO) were also set up. According to the absorbance, the inhibition rate of cell proliferation in different time periods was analyzed.

### 2.7. In Vitro Cell Migration Assays

After being seeded in six-well plates at a density of 3 × 10^5^ cells/well, A549 cells were allowed to adhere to the well bottoms and incubate for 24 h. Straight trauma lines were created with a 200 μL pipette tip, followed by three washes with PBS to remove floating cells. Then, 1% FBS was used to incubated the cultured cells under the condition of 5% CO_2_ and 37 °C. 1% DMSO was used as the control group and compounds 1, 6 and 7 with a concentration of 200 μM as the experimental groups, and the migration was sequentially photographed at 0, 6, 12 and 24 h. Equation (1) below reveals the calculation of the cell migration rate in each group:Migration rate = (W_a_/W_c_) × 100%(1)
where W_a_ and W_c_ refer to the migration distance of the experimental group and the control group, respectively.

### 2.8. Cell Apoptosis Assays

Apoptotic morphology was observed using Hoechst 33258 staining. A549 cells were suspended and inoculated in six-well plates at a density of 3 × 10^5^ cells/well. The cells were separately plastered with each of three compounds (200 μM) and after 24 h, the culture medium was removed and fixed in 4% formaldehyde for 15 min. The cells were washed three times with PBS to remove the formaldehyde. Finally, the Hoechst 33258 working solution was added and stained for 10 min, then washed 3 times with PBS and sealed. Apoptosis was detected on a flow cytometer using the AnnexinV-FITC/PI Apoptosis Assay Kit. Cells were digested and suspended using EDTA-free trypsin digest, centrifuged at 1800 r/min for 10 min, removed from the culture, resuspended in PBS and centrifuged again for three times. Then, 500 μL of binding buffer was added to suspend the cells, followed by 5 μL of Annexin V-EGFP and 5 μL of Propidium Iodide(PI) to the drug group. The reaction was carried out for 10–20 min at room temperature, protected from light and detected by flow cytometry.

### 2.9. Flow Cytometry Analysis of Cell Cycle

The A549 cells in logarithmic growth phase were co-incubated with each of three compounds at 5% CO_2_ and 37 °C for 24 h, and then, the culture medium was removed and digested with trypsin and the cells were suspended by blowing the culture medium. After centrifugation at 800 rpm for 15 min, the supernatant was removed and the cells were evenly suspended again with PBS. The cells were washed and centrifuged twice. The cells were resuspended with 0.4 mL PBS and transferred to tube for gentle blowing so as to prevent the cell breakage. About 3% RNase-A was added to reach a final concentration of about 50 μg/mL for the digestion for 30 min at a 37 °C water bath. An amount of 50 μL PI solution was then added to reach a final concentration of about 65 μg/mL in ice bath for 30 min. Finally, 300 mesh (aperture 40 ~ 50 μm) nylon net was used to filter the solution followed by testing with the flow cytometer.

## 3. Results and Discussion

### 3.1. Virtual Screening Results

A total of 208332 candidate molecules were selected in SPECS database for the virtual screening. The 3D similarity between the candidate compounds and the six query molecules in the database was calculated by SHAFTS, an efficient virtual screening tool, which not only considers the similarity of 3D structure but also takes the similarity between candidate molecules and pharmacophores in the template into account. Compared with the traditional shape-based 3D molecular similarity screening program ROCS, SHAFTS is superior to some other similar methods in calculation accuracy and was successfully applied to the discovery of RSK2 inhibitors [17]. Therefore, it is considered as a more efficient calculation method. In the calculation process, the similarity threshold was set as 1.0, and 4012 eligible compounds were preliminarily screened as the input of the next docking experiment. The coefficient of the feature score set at about 1.0 can ensure that the shape score and the feature score have the same weight to determine the final hybrid score. In most cases, setting the weight factor to 1.0 gives better results than other values 0.5, 0.8, 1.2 and 1.5 [18].

After preliminary screening, in order to find effective G-quadruplex ligands accurately, the molecular docking experiments were carried out as further screening. According to the different conformations of G-quadruplex, four typical structures were selected as targets with parallel structure, antiparallel structure or a combination of both. The binding modes of the selected compounds with G-quadruplex were shown in Appendix A. Small molecules with different conformations were mainly stacked at the bottom or top of G-tetrad and inserted in the big groove. The Poisson–Boltzmann surface area method was used to express the binding score between receptor and ligand molecules [29,30]. The smaller the docking score is, the more reliable the combination between the candidate compounds and G-quadruplex becomes [31,32]. According to the sequence of docking score, 23 molecules were selected as hit compounds. Their chemical formulas were shown in Appendix A and Appendix A. The docking scores of the target G-quadruplex with ligands were shown in Appendix A. The results indicates that these compounds could stably bind to G-quadruplexes with four different conformations (Appendix A) since the binding scores were mostly between −6 and −5. In addition, the 3D similarity scores of the 23 hit compounds were listed in Appendix A. Finally, compounds 1, 6 and 7 of 23 hit compounds were selected for further verification experiments. Table 1 and Table 2 show the docking scores and 3D similarity calculation scores of the three compounds 1, 6 and 7 accompanied with their binding to the four conformations of G- quadruplexes (143D, 1KF1, 2HY9 and 2JPZ), as shown in Figure 3.

### 3.2. Cytotoxicity of the Compounds

In order to study the anticancer activity of the selected hit compounds, their cytotoxicity toward the lung cancer cells A549 were investigated individually with the MTT viability assay through treating cells with the compounds at different concentrations (0.78–400 μM) for 24 h. The calculated IC_50_ values of all the compounds were listed in Table 3. Clearly, compounds 1, 6 and 7 showed the best cytotoxic effect on A549. As compared with that of drug molecules known to target the G-quadruplex and inhibit cell proliferation [33], the obtained cytotoxicity data were found to be comparable, further validating the accuracy of the virtual screen. Compounds 1, 6 and 7 were, thus, selected for the subsequent cellular experiments.

### 3.3. In Vitro Inhibition Behavior

MTT assay was used to detect the cytotoxicity of the selected compounds 1, 6 and 7 at 200 μM towards A549 cells. As shown in Figure 4, compounds 1 and 6 did not show a good ability to inhibit cell proliferation in a 24 h drug treatment but displayed a certain inhibitory effect over time. Afterwards, the inhibition ability gradually weakened and the tumor cells gradually returned to proliferation. Among the three compounds, compound 7 displayed the best inhibitory effect on the proliferation of tumor cells but no time-dependent effect. Therefore, the experimental results showed that compounds 1 and 6 could inhibit tumor cells to a certain extent, while compound 7 was capable of effectively inhibiting tumor cells.

### 3.4. Investigations of In Vitro Migration

Cell scratch method is to create a “wound” in the cell monolayer and evaluate the effect of drugs on cell migration in vitro by co-incubating cell with drug [34]. The growth of cells and their forward movement to the central scratch area mean the existence of cell migration. Figure 5 and Figure 6 showed the time-dependent migration of A549 cells to scratch area in control group and groups treated with compounds 1, 6 and 7, respectively. After a 24 h of treatment, the scratch of the control group was basically closed, and the cell migration rate was maintained over 90%. However, after being treated with each of compounds 1, 6 and 7 for 24 h, A549 cells did not obviously migrate to the scratch area and the cell migration rate remained between 11% and 20%. These results indicates that the hit compounds selected could effectively inhibit the migration of A549 cells and might weaken the tumor migration in vitro. As compounds 1, 6 and 7 entered cancer cells, their pharmacological effects significantly inhibited the activity of A549 cells. As a consequence, no obvious cell migration to scratch area was observed in groups treated with compounds.

### 3.5. Analysis of Apoptosis

Apoptosis involves the activation, expression and regulation of genes, and is a process of active cell death with complex mechanisms [35,36]. As shown in Figure 7, regions Q1, Q2, Q3 and Q4 refer to (AnnexinV-FITC)-/PI+, (AnnexinV-FITC)+/PI+, (AnnexinV-FITC)+/PI- and (AnnexinV- FITC)-/PI-, indicating the proportions of necrotic cells, late apoptotic cells, early apoptotic cells and live cells, respectively. As seen from the results of annexin-V FITC/PI double staining, in the control group, only a small amount of A549 cells displayed apoptosis after 24 h (the upper right quadrant was late apoptotic cells, and the lower right quadrant the early apoptotic cells). In contrast, the compounds selected by virtual screening could induce the obvious apoptosis of A549 cells. Figure 7C shows that compound 6 induced 20.47% of apoptosis (Q_2_ + Q_3_), while compound 7 induced a higher apoptosis of 27.95% (Figure 7D, Q_2_ + Q_3_), which was significantly better than that of the control. Moreover, the Hoechst staining assays further confirmed the significant abilities of compounds 1, 6 and 7 to kill A549 cancer cells (Figure 8). Normal cells in the control group were only faintly fluorescent. The majority of cells treated with compounds 1, 6 and 7 were significantly condensed and exhibited highly fluorescent nuclei, a characteristic pattern of apoptosis. In particular, a clear crescent-shaped apoptosis could be observed in the compound 7-treated group. Compared with the previously reported data [25,36], these results suggest that the small molecules targeting G-quadruplexes could stabilize the existence of G-quadruplexes and remarkably inhibit the telomerase activity in tumor cells, which may start the apoptosis pathway of tumor cells and induce apoptosis.

### 3.6. The Result of Cell Cycle

Cell cycle refers to the whole process of a cell from the completion of one division to the end of the next division. The genetic material of a cell replicates and equally distributes to two daughter cells. Therefore, the content of DNA in cells changes with the progress of cell cycle [37,38]. PI dye is used to label and the relative content of DNA in cells is determined by flow cytometry. The percentage of each phase of cell cycle can be analyzed, and then, the effect of drugs on cell cycle can be determined. As seen from Figure 9, G1 phase of compounds 1, 6 and 7 increased significantly, while S phase and G2 phase decreased, especially S phase was shortened by more than a quarter. In theory, when the cell is damaged, it cannot complete the cell cycle normally and will be blocked at a certain stage, triggering a series of repair reactions. If the repair is good, the cell cycle will continue, if not, it will trigger the apoptosis mechanism for apoptosis. Therefore, based on the experimental results, the cell cycle of A549 cells was arrested in G1 phase, and the replication of DNA was relatively reduced. Therefore, the selected compounds could effectively inhibit the division process of tumor cells.

## 4. Conclusions

G-quadruplexes was successfully used as a potential target for drug virtual screening so as to find effective anticancer drugs more accurately. A total of 23 compounds were finally hit based on the 3D similarity and docking score results. The docking experiment results showed that the hit compounds could interact by stacking on the G-quadruplexes plane, spreading in the small or large DNA groove, or directly inserting into it. All the hit compounds had planar aromatic rings with generally protonated branched chains and were thus charged, which could form a stable stacking structure. Furthermore, compounds 1, 6 and 7 were selected for the subsequent cell experiments in vitro in order to further verify the anticancer activity of the hit compounds. The tests of both cell proliferation inhibition and scratch proved that the three compounds could effectively inhibit the proliferation and migration of A549 cells, and compound 7 did the best. This is in good accordance with the previously reported results that small molecules targeting G-quadruplexes ligand can stabilize the existence of G-quadruplexes for the purpose to inhibit the telomerase activity inside tumor cells, thereby inhibiting the proliferation of tumor cells. The flow cytometry analysis further confirmed that the compound could induce the apoptotic pathway of tumor cells and finally cause the apoptosis of tumor cells. The results of cell cycle further verified that the existence of these small molecules targeting G-quadruplexes could make A549 cell cycle stay in G1 phase, hindering both DNA replication and cell division.

In conclusion, these findings revealed that the compounds based on the virtual screening may provide a new way for new drug design targeting G-quadruplexes, thus stimulating further research on exploring more effective anticancer drugs. Furthermore, the selected compounds especially compound 7 proved to exhibit favorable anticancer activity and, therefore, is worthy of further study.

## Figures and Tables

**Figure 1 pharmaceutics-15-01414-f001:**
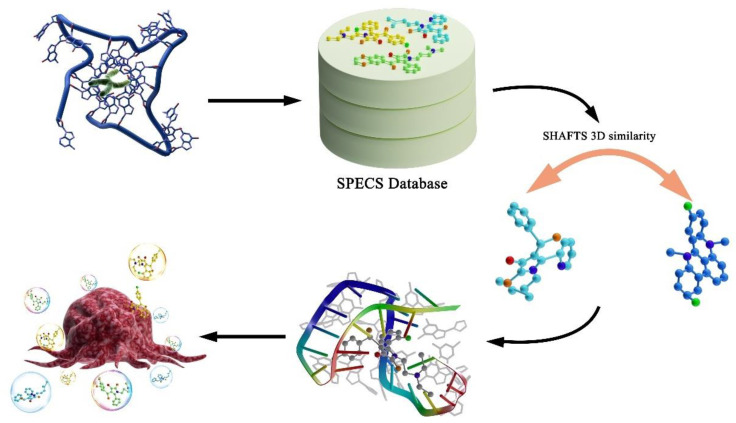
The figure shows the overall work flow. The purpose of the study is to screen out potential drugs targeting G-quadruplex. The first step is to select the SPECS database as the candidate molecular database. The second step is to calculate the three-dimensional similarity of candidate molecules according to query molecules. Additionally, we further filter the potential effective drugs through molecular docking experiments. Finally, we selected compounds 1, 6 and 7 according to the similarity score and docking score to carry out cell inhibition experiment, cell migration assays, cell apoptosis assay and cell cycle analysis to verify that the drugs obtained by virtual screening do have anticancer activity.

**Figure 2 pharmaceutics-15-01414-f002:**
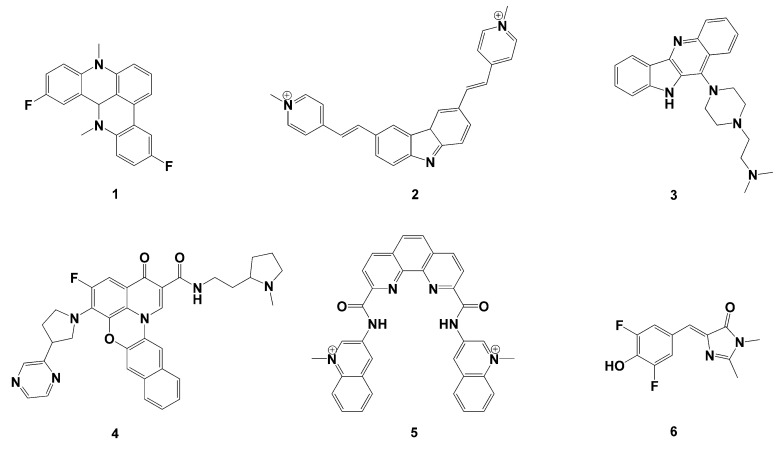
The chemical structures of six query molecules used as the template to calculate 3D similarity.

**Figure 3 pharmaceutics-15-01414-f003:**
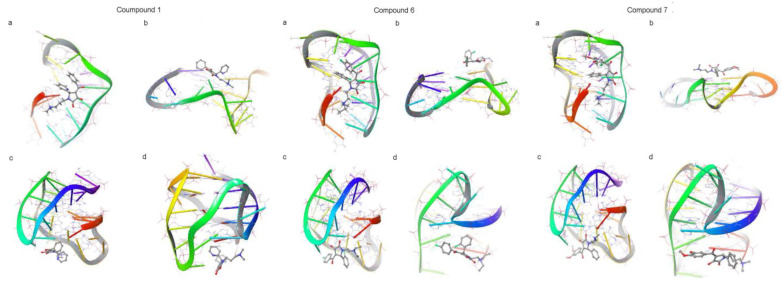
Four conformations of G-quadruplexes binding with the three screened compounds 1, 6 and 7 ((**a**–**d**): 143D, 1KF1, 2HY9, 2JPZ).

**Figure 4 pharmaceutics-15-01414-f004:**
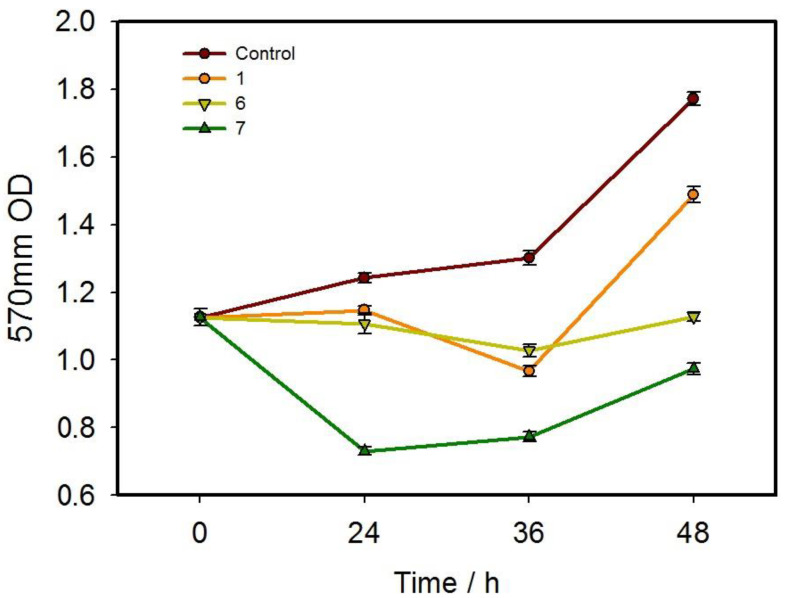
The inhibitory effect of the compounds 1, 6 and 7 at 200 μM on the proliferation of human lung cancer cell line A549 at different time (using DMSO as control group).

**Figure 5 pharmaceutics-15-01414-f005:**
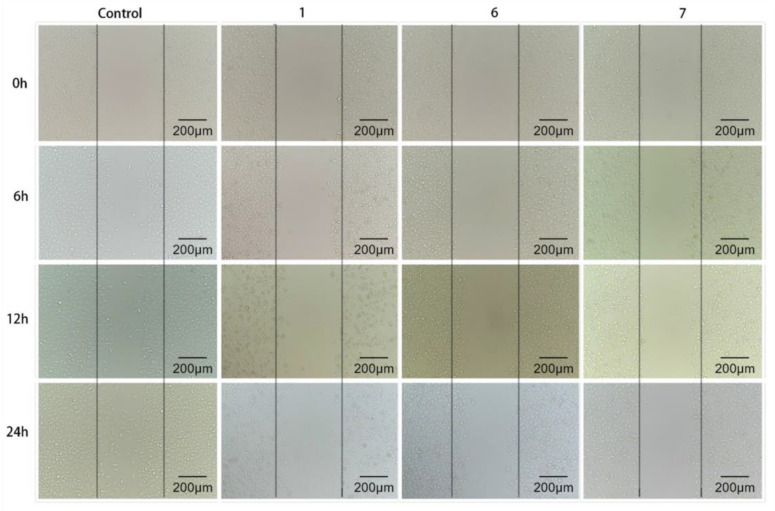
The migration evaluation of A549 cell motility in both control and groups treated with compounds 1, 6 and 7.

**Figure 6 pharmaceutics-15-01414-f006:**
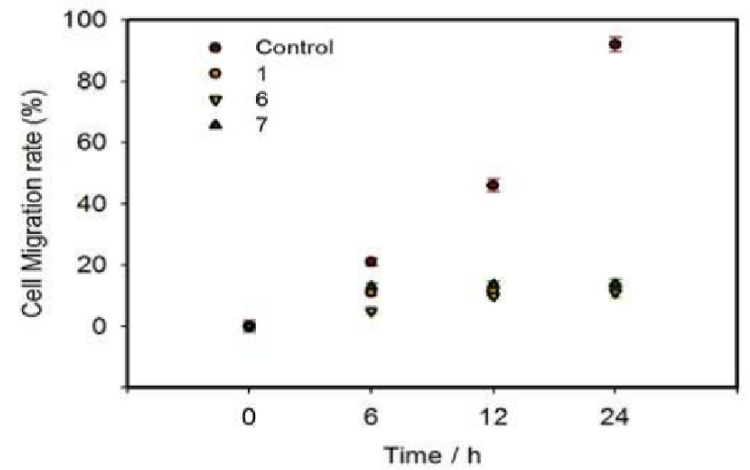
The cell migration rate after treatment with the three compounds.

**Figure 7 pharmaceutics-15-01414-f007:**
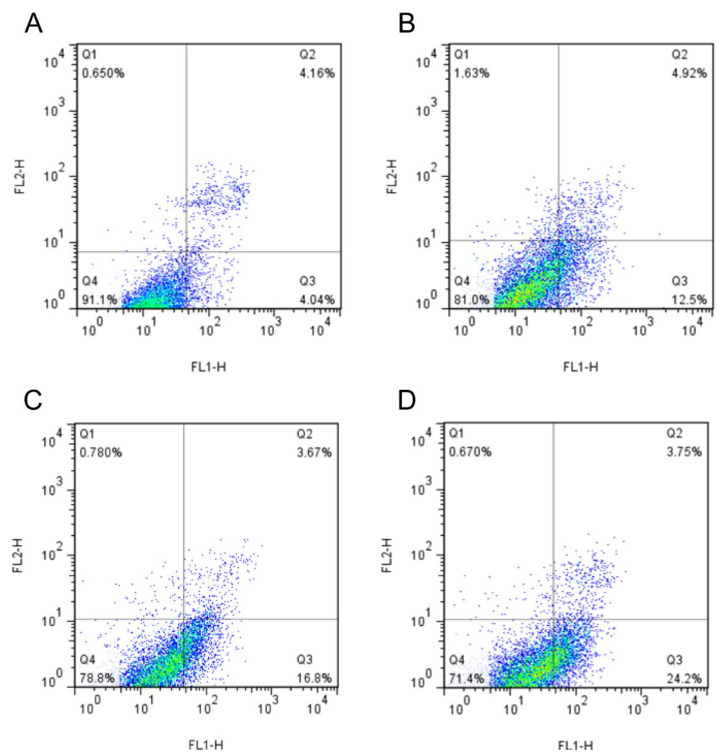
Flow cytometry analysis for apoptosis of A549 cell in control group ((**A**), DMSO) and groups treated with 200 μM compounds 1 (**B**), 6 (**C**) and 7 (**D**) after 24 h, respectively, for 24 h. The cells were harvested, stained with Annexin V-FITC and PI, and analyzed by flow cytometry. FL1: the green channel of Annexin-V−FITC; FL2: the red channel of PI.

**Figure 8 pharmaceutics-15-01414-f008:**
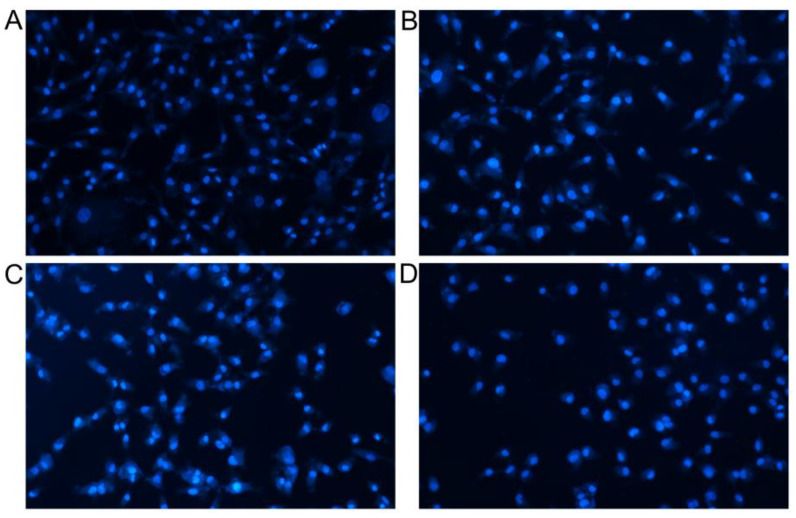
Micrographs of nuclear staining of cell monolayer for control group (**A**) and groups treated with compounds 1 (**B**), 6 (**C**) and 7 (**D**) by Hoechst staining.

**Figure 9 pharmaceutics-15-01414-f009:**
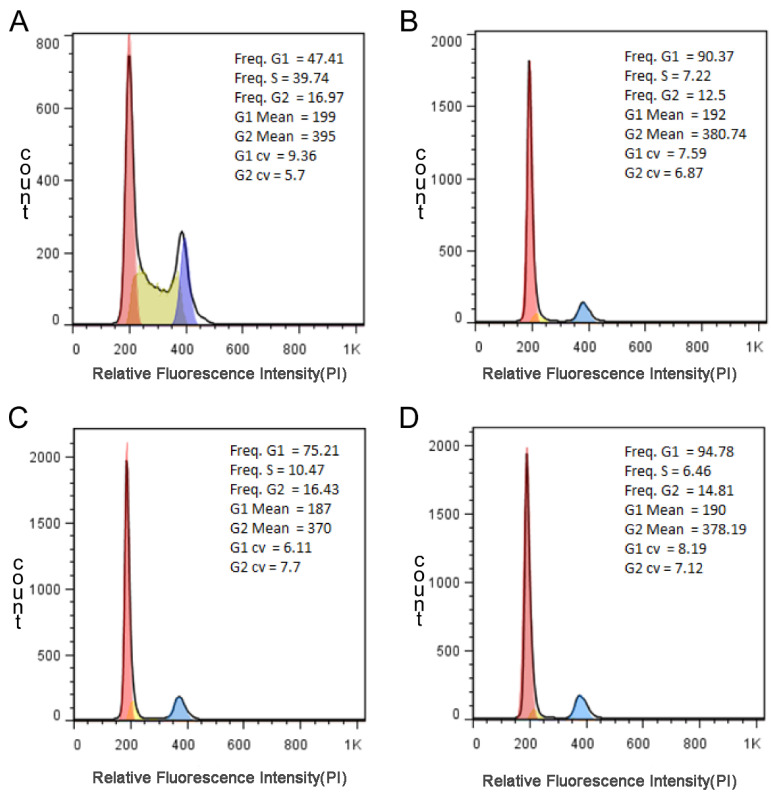
Cell cycle effect of DMSO ((**A**), control group), compounds 1 (**B**), 6 (**C**) and 7 (**D**) at 200 μM on A549 cells. The cells in logarithmic growth phase were co-incubated with three compounds at 5% CO_2_ and 37 ^o^C for 24 h. The cells were trypsinized, harvested and washed three times with ice-PBS for PI-stained DNA content detected by flow cytometry.3% RNase-A (50 μg/mL) in water bath at 37 ^o^C for 30 min; PI (65 μg/mL) in ice bath for 30 min.

**Table 1 pharmaceutics-15-01414-t001:** Docking scores of compounds 1, 6 and 7.

Number		Docking Score		
	143D	1KF1	2HY9	2JPZ
1	−6.36921	−3.30817	−5.4937	−3.87285
6	−5.01954	−2.71853	−5.00908	−4.88085
7	−5.01123	−3.61745	−4.81507	−4.20793

**Table 2 pharmaceutics-15-01414-t002:** 3D similarity calculation scores of compounds 1, 6 and 7.

Number	Template	Feature Score	Shape Score	Hybrid Score
1	1	0.44	0.785	1.227
6	1	0.521	0.729	1.25
7	3	0.273	0.676	0.95

**Table 3 pharmaceutics-15-01414-t003:** Cytotoxicity of 23 hit compounds.

Compounds	1	2	3	4	5
IC_50_ ± SEM (μΜ/L)	201.35 ± 1.2	259.64 ± 2.6	345.75 ± 3.8	318.64 ± 4.1	298.27 ± 1.8
Compounds	6	7	8	9	10
IC_50_ ± SEM (μΜ/L)	198.38 ± 2.3	185.89 ± 1.7	314.19 ± 3.1	245.74 ± 0.87	349.87 ± 5.6
Compounds	11	12	13	14	15
IC_50_ ± SEM (μΜ/L)	326.48 ± 4.6	>400	>400	396.54 ± 4.8	>400
Compounds	16	17	18	19	20
IC_50_ ± SEM (μΜ/L)	390.23 ± 2.8	>400	>400	374.56 ± 5.2	>400
Compounds	21	22	23		
IC_50_ ± SEM (μΜ/L)	>400	389.65 ± 3.5	>400		

## Data Availability

Not applicable.

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
