# Peer review of "Virtual Screening-Based Study of Novel Anti-Cancer Drugs Targeting G-Quadruplex"

_pharmaceutics, 2023, doi:10.3390/pharmaceutics15051414_

Round 1
Reviewer 1 Report
Ouyang et al. present a computational screening approach to selecting candidate molecular pharmaceuticals to specifically bind to the G quadruplex secondary structure of DNA. As the authors explain, G quadruplexes are thought to prevail in cancer cells and thus serve as a candidate for therapeutic treatment. The authors screen through over 200,000 compounds from a database and choose 23 compounds which then undergo various similarity calculations (to the target) and molecular docking simulations. Three compounds are chosen for select in vitro studies to assess, for example, apoptosis. The premise for this study using a virtual platform as a screening aide is worthy of pursuit since experimental drug screening is expensive and time consuming. However, this reviewer was puzzled by some of the wording choices of the authors, some of experimental data presented in the Figures, and the Figure captions which appeared incomplete in their description of data presented.
Throughout the manuscript, there were some odd, occasionally meandering phrasing choices or sentence structures. While not scientifically detrimental, clearer wording choices would help the reader focus on the scientific narrative. As one example, sentences in the first paragraph (lines 31-35) reads “But these drugs are not specific and inevitably have serious side effects, from which the human body may greatly suffer. In recent years, as more research that related to tumor signaling pathways were further studied, many new anti-cancer drugs have been designed by using protein kinases or surface receptors as specific targets crucial to reduce drug side effects and improve drug efficacy, so it is necessary to explore specific drugs for effective targets.” This reviewer suggests a more concise approach as follows. “These drugs are not sufficiently target specific and inevitably have serious side effects. In recent years as more research on tumor signaling pathways emerged, many new anti-cancer drugs have been designed using protein kinases or cell membrane receptors as specific targets in order to simultaneously reduce drug side effects and improve drug efficacy.”
Other wording choices, though infrequent, appeared more problematic. As an example of possibly misleading wording choices, the authors describe a G-quadruplex as a “four-stranded secondary structure.” (line 37) As this reviewer understands it, however, G-quadruplexes involve hydrogen bonding between 4 guanine bases on the same strand, not four distinct oligonucleotide strands. Other minor, but potentially confusing, mistakes in wording should be corrected throughout the Materials and Methods section. For example, what do the authors mean in line 158 that “the cell monolayer was covered with flat bottom”? In line 180, it appears that the authors reversed their definition of Wa (migration distance of experimental group) and Wc (migration distance of control group) in their written description. In line 183, the authors mention studying “apoptosis morphology”; however, it appears that the authors were actually using flow cytometry to assess apoptosis, not cell shape. It is unclear to this reviewer what the phrase “and then tested on the computer” refers to in line 206.
The Figure captions for Figures 3, 6, and 7 need to be heavily revised to describe the data presented in sufficient and accurate detail. Discussion of the interpretation of the Figure’s data should be reserved for the discussion section in the manuscript. One example illustrating this critique is Figure 7 which appears to shows 4 flow cytometry histograms, but with a confusing x-axis labeled “DNA content” with numerical values, but no units (e.g., micrograms of DNA?). Flow cytometry measures fluorescence intensity of a (sub)population of gated cells, not DNA content. As this reviewer understands this experiment, the authors are measuring the amount of labeled DNA (which decreases by 50% every time a cell divides into two daughter cells), not the amount of total DNA. For Figure 6 and the discussion in Section 3.4, it is not clear what apoptosis “rate” (e.g., a measure of how quickly cells die?) refers in line 287 and how the percentages reported in line 288 (20.47% in (Figure 6C)…27.9% (Figure 6D)) were calculated. The authors describe the two right quadrants as representing apoptotic cells, but their approach for calculating apoptosis “rate” was not clear.
Figure 4 was the most problematic for this reviewer in terms of the data itself. Assuming these are micrographs with no scale bar, this reviewer was unable to see any cells. Finally, the caption for "Figure S2" (need to renumber as Figure S1) in the Supporting Information needs to add information about A, B, C, D for each compound simulation (e.g., 4 specific types of G quadruplex secondary structures binding to a particular molecular compound?)
While this reviewer understands that computational data is important, it was difficult to appreciate Table 1 and Table 2 and aspects of the discussion in lines 223-238. For example, the authors state that “the smaller the docking binding energy, the more reliable the combination between the candidate compounds and G-quadruplex” (lines 231-232). However, isn’t a system more energetically favorable as its binding free energy becomes more negative rather than smaller (in magnitude)? In lines 236-237, the authors state that “the binding energies are mostly between -6 and -5, but no units (e.g., kcal/mol) are provided in this sentence. More confusing is that Table 1 lists “docking scores”, not binding energies. Tables simply listing data for multiple compounds are often hard for an outside reader to appreciate and these inconsistencies left this reviewer a bit confused. Perhaps slimming these tables to present only the 3 compounds experimentally verified accompanied by select 3D models from “Figure S2” in Supporting Information (line 234 though only one multi-panel Figure was found in Supporting Information) would be a better choice of space for the main manuscript. The complete data sets for Tables 1 &2, currently in the main manuscript, could then be moved to Supporting Information.
While these suggestions involve substantive editing by the authors, these suggested changes are intended to allow a future reader to better appreciate the potential impact of the computational screening approaches to improving drug selection for cancer treatment.
Author Response
1. Throughout the manuscript, there were some odd, occasionally meandering phrasing choices or sentence structures. While not scientifically detrimental, clearer wording choices would help the reader focus on the scientific narrative. As one example, sentences in the first paragraph (lines 31-35) reads “But these drugs are not specific and inevitably have serious side effects, from which the human body may greatly suffer. In recent years, as more research that related to tumor signaling pathways were further studied, many new anti-cancer drugs have been designed by using protein kinases or surface receptors as specific targets crucial to reduce drug side effects and improve drug efficacy, so it is necessary to explore specific drugs for effective targets.” This reviewer suggests a more concise approach as follows. “These drugs are not sufficiently target specific and inevitably have serious side effects. In recent years as more research on tumor signaling pathways emerged, many new anti-cancer drugs have been designed using protein kinases or cell membrane receptors as specific targets in order to simultaneously reduce drug side effects and improve drug efficacy.”
Reply: According to your valuable suggestions, the sentence was revised in the revision. (see lines 31-35)
2. Other wording choices, though infrequent, appeared more problematic. As an example of possibly misleading wording choices, the authors describe a G-quadruplex as a “four-stranded secondary structure.” (line 37) As this reviewer understands it, however, G-quadruplexes involve hydrogen bonding between 4 guanine bases on the same strand, not four distinct oligonucleotide strands. Other minor, but potentially confusing, mistakes in wording should be corrected throughout the Materials and Methods section. For example, what do the authors mean in line 158 that “the cell monolayer was covered with flat bottom”? In line 180, it appears that the authors reversed their definition of Wa (migration distance of experimental group) and Wc (migration distance of control group) in their written description. In line 183, the authors mention studying “apoptosis morphology”; however, it appears that the authors were actually using flow cytometry to assess apoptosis, not cell shape. It is unclear to this reviewer what the phrase “and then tested on the computer” refers to in line 206.
Reply: We greatly appreciate your valuable suggestion. The abovementioned issues were addressed in the revised manuscript. (see lines 36-37, 181-186, 223-224, 249-250, and section 3.5)
3. The Figure captions for Figures 3, 6, and 7 need to be heavily revised to describe the data presented in sufficient and accurate detail. Discussion of the interpretation of the Figure’s data should be reserved for the discussion section in the manuscript. One example illustrating this critique is Figure 7 which appears to shows 4 flow cytometry histograms, but with a confusing x-axis labeled “DNA content” with numerical values, but no units (e.g., micrograms of DNA?). Flow cytometry measures fluorescence intensity of a (sub)population of gated cells, not DNA content. As this reviewer understands this experiment, the authors are measuring the amount of labeled DNA (which decreases by 50% every time a cell divides into two daughter cells), not the amount of total DNA. For Figure 6 and the discussion in Section 3.4, it is not clear what apoptosis “rate” (e.g., a measure of how quickly cells die?) refers in line 287 and how the percentages reported in line 288 (20.47% in (Figure 6C)…27.9% (Figure 6D)) were calculated. The authors describe the two right quadrants as representing apoptotic cells, but their approach for calculating apoptosis “rate” was not clear.
Reply: Your valuable comments are greatly appreciated. As suggested, the mentioned descriptions were well revised. (see figures 4, 7, and 9, lines 325-326, 370-372, 393-396, and section 3.5)
4. Figure 4 was the most problematic for this reviewer in terms of the data itself. Assuming these are micrographs with no scale bar, this reviewer was unable to see any cells. Finally, the caption for "Figure S2" (need to renumber as Figure S1) in the Supporting Information needs to add information about A, B, C, D for each compound simulation (e.g., 4 specific types of G quadruplex secondary structures binding to a particular molecular compound?)
Reply: According to your meaningful suggestions, the scale bar was added in Figure 4. In the ESI, Figure S2 was not changed as Figure S1 was added. We greatly appreciated your valuable suggestion on adding information about a, b, c, d for each compound simulation. In this work, four different conformations of G-quadruplex (a-b:143D, 1KF1, 2HY9, 2JPZ) were selected for the binding between G-quadruplex with each of the 23 hit compounds, which were clarified in the caption of Figure S2. (see Figure 5, and Figure S2)
5. While this reviewer understands that computational data is important, it was difficult to appreciate Table 1 and Table 2 and aspects of the discussion in lines 223-238. For example, the authors state that “the smaller the docking binding energy, the more reliable the combination between the candidate compounds and G-quadruplex” (lines 231-232). However, isn’t a system more energetically favorable as its binding free energy becomes more negative rather than smaller (in magnitude)? In lines 236-237, the authors state that “the binding energies are mostly between -6 and -5, but no units (e.g., kcal/mol) are provided in this sentence. More confusing is that Table 1 lists “docking scores”, not binding energies. Tables simply listing data for multiple compounds are often hard for an outside reader to appreciate and these inconsistencies left this reviewer a bit confused. Perhaps slimming these tables to present only the 3 compounds experimentally verified accompanied by select 3D models from “Figure S2” in Supporting Information (line 234 though only one multi-panel Figure was found in Supporting Information) would be a better choice of space for the main manuscript. The complete data sets for Tables 1 &2, currently in the main manuscript, could then be moved to Supporting Information.
Reply: According to your meaningful and valuable suggestions, the relevant descriptions were revised to make them readable and understandable. Also, the complete data sets for Tables 1 &2 were moved to ESI. (see lines 275-286, Tables S2-3 in the ESI)
Reviewer 2 Report
The manuscript entitled “Virtual Screening-based Study of Novel Anti-cancer Drugs Targeting G-quadruplex” by Ruizhuo Ouyanga, et al. used virtual screening to identify potential small molecule drug candidates that target G-quadruplexes and evaluate their anti-cancer activity through in vitro experiments. The authors employed the SHAFTS method to do 3D molecular similarity calculations, using six query molecules as a basis. The screened drug candidates were further evaluated by docking experiments which resulted in 23 hit molecules. In vitro experiments including MTT, cell migration, apoptosis, and cell cycle analysis were performed to study the anti-cancer activity of the three selected molecules. The study design, method, experiments, and data analysis are well described. However, the work has some flaws that needed to be addressed. The overall presence of the current manuscript is too early for publication. Key issues are listed below.
1. In the Introduction section, the background and significance of virtual screening are not described sufficiently. Please revise.
2. Figure 1 and its legend is too concise. The authors should at least provide key descriptions in each step and in the legend to present the workflow more clearer.
3. It’s unclear how the four G-quadruplex 143D, 1KF1, 2HY9, 2JPZ were chosen. What’s the rationale behind this?
4. The chemical formulas of 23 hit molecules are missing in table S1. What are their chemical structures?
5. The synthesis of compounds 1, 6, and 7 are not described in this work. Please clarify. What is their solubility and how were the drug solutions prepared?
6. What’s the rationale for why compounds 1, 6, and 7 were chosen? What’s the difference between these three? Though the authors state that compounds 1, 6, and 7 shows good similarity scores and docking scores, there are some others in the 23 hits show very similar scores. Please clarify.
7. Why did the authors only select three molecules into the in vitro experiments? It’s expected that more molecules should be evaluated in cell experiments to better validate the virtual screening,
8. To confirm the accuracy of the virtual screening and understand better the drug efficacy of the selected molecules, the authors should include a positive control (e.g. a known drug molecule that can target G-quadruplexes and inhibit cell proliferation) and a negative control (e.g. a molecule with low scores from the current virtual screening) in the cell experiments for comparison.
9. Besides the docking score, the authors should include experimental data to show the binding kinetics of the G-quadruplexes and their ligands.
10. The cell experiments demonstrate the three selected compounds show tumor cells inhibitory effect, but the results and data cannot support the conclusion that those anti-cancer activities were attributed to G-quadruplexes targeting and telomerase inhibition. Please discuss and add experimental data to support this.
11. Please add the full description for the abbreviation of physical properties in table S1.
12. Some typos were found in the manuscript. For example, on page 2 line 56 “small molecular”. On page 6 line 234, figure S2?
Author Response
1. In the Introduction section, the background and significance of virtual screening are not described sufficiently. Please revise.
Reply: According to your valuable suggestion, the background and significance of virtual screening was revised. (see Introduction section)
2. Figure 1 and its legend is too concise. The authors should at least provide key descriptions in each step and in the legend to present the workflow more clearer.
Reply: Thanks a lot for your valuable comments. As suggested, the legend of Figure 1 was revised. (see figure 1)
3. It’s unclear how the four G-quadruplex 143D, 1KF1, 2HY9, 2JPZ were chosen. What’s the rationale behind this?
Reply: We are very grateful for your valuable suggestions.The reasons for the choice of the four G-quadruplex was provided in the revision. (see lines 169-173)
4. The chemical formulas of 23 hit molecules are missing in table S1. What are their chemical structures?
Reply: Your valuable suggestions are greatly appreciated. The chemical formulas of 23 hit molecules were included in the electronic supporting information (ESI). (see Figure S1 in the ESI)
5. The synthesis of compounds 1, 6, and 7 are not described in this work. Please clarify. What is their solubility and how were the drug solutions prepared?
Reply: Thanks a lot for your nice comments. After being screened out, compounds 1, 6 and 7 were purchased from the reagent company. These three compounds were dispersed in DMED culture broth for use. (see section 2.5)
6. What’s the rationale for why compounds 1, 6, and 7 were chosen? What’s the difference between these three? Though the authors state that compounds 1, 6, and 7 shows good similarity scores and docking scores, there are some others in the 23 hits show very similar scores. Please clarify.
Reply: Thank you very much for your meaningful questions. Besides compounds 1, 6, and 7 shows good similarity scores and docking scores, lower IC50 values were found for these three compounds than other 20 compounds. Their structures of the main backbones are very similar as well. So compounds 1, 6, and 7 were chosen. (see section 3.2)
7. Why did the authors only select three molecules into the in vitro experiments? It’s expected that more molecules should be evaluated in cell experiments to better validate the virtual screening.
Reply: Your valuable suggestions are greatly appreciated. After comparing the cytotoxicity of these 23 compounds, compounds 1,6 and 7 were found to be more effective and therefore were chosen for the subsequent experiments. (see section 3.5)
8. To confirm the accuracy of the virtual screening and understand better the drug efficacy of the selected molecules, the authors should include a positive control (e.g. a known drug molecule that can target G-quadruplexes and inhibit cell proliferation) and a negative control (e.g. a molecule with low scores from the current virtual screening) in the cell experiments for comparison.
Reply: According to your valuable suggestions, the drug molecules known to target the G-quadruplex and inhibit cell proliferation were selected to to confirm the accuracy of the virtual screening in the cytotoxicity experiment for comparison. (see section 3.2)
9. Besides the docking score, the authors should include experimental data to show the binding kinetics of the G-quadruplexes and their ligands.
Reply: We greatly appreciated your valuable suggestions and totally agree with your comments. However, the experiments to show the binding kinetics of the G-quadruplexes and their ligands could not be performed in the current platform of our lab. We apologize for it.
10. The cell experiments demonstrate the three selected compounds show tumor cells inhibitory effect, but the results and data cannot support the conclusion that those anti-cancer activities were attributed to G-quadruplexes targeting and telomerase inhibition. Please discuss and add experimental data to support this.
Reply: Your valuable suggestions are greatly appreciated. Based on the previously reported results and our screening and experimental data, the anticancer activity of these three compounds may be attributable to G-quadrilateral targeting and telomerase inhibition. However, due to the limited conditions of our lab, the relevant experiments could not be carried out as suggested. We will make our great efforts to achieve these experimental data in our upcoming work. We are so sorry about this. (see section 3.5)
11. Please add the full description for the abbreviation of physical properties in table S1.
Reply: According to your valuable suggestions, the full description for the abbreviation of physical properties were added in Table S1. (see Table S1 in ESI)
12. Some typos were found in the manuscript. For example, on page 2 line 56 “small molecular”. On page 6 line 234, figure S2?
Reply: Thank you very much for pointing out the mistakes, which were corrected in the revision. Also, the whole text was carefully checked to improve the quality of this paper.
Round 2
Reviewer 1 Report
Overall, the authors address nearly all of this reviewer’s concerns and suggestions. The effort to make these changes as well as to annotate these changes in both the revised manuscript as well as the response letter was clearly evident and thus appreciated by the reviewer.
The language and grammar are much improved, but this reviewer suggests that the authors get a fresh pair of eyes to help fix minor residual grammatical errors throughout the manuscript. A few suggestions are included below for p. 1 only below.
Change “hydrogen bond” to “hydrogen bonds” in line 37
Change “synlanti” to “syn and anti” in line 39
Change “environment transitions” to “environmental conditions” in line 40
Change “According to the different direction of nucleotide chain, it can be divided into” to “Depending on the 5’-3’ versus 3’-5’ orientation of segments in the self-folded oligonucleotide chain(s), G quadruplexes can be categorized as” in lines 41-42.
The addition of Figure 8 is meaningful, but this reviewer does suggest that the wording in the Figure caption should be revised to simply state a brief description of the data (e.g., Figure 8. Micrographs of nuclear staining of cell monolayer for control group (A)…), not a discussion of what can be inferred or concluded from the experiment (e.g., exclude the phrase “Detection of apoptosis”)
Figure 9 is the only example in which the authors did not appear to make a suggested change from this reviewer’s previous comment. Since they also did not explain why, this reviewer is still of the opinion that the x-axis in Figure 9 (renumbered in revised manuscript) should be labeled as “Relative Fluorescence Intensity” rather than with a unitless “DNA content” label.
Author Response
1. Change “hydrogen bond” to “hydrogen bonds” in line 37.
Reply: According to your valuable suggestion, the words were revised in the revision. (see lines 37)
2. Change “synlanti” to “syn and anti” in line 39.
Reply: We greatly appreciate your valuable suggestion, the words were revised in the revision. (see lines 39)
3. Change “environment transitions” to “environmental conditions” in line 40.
Reply: Your valuable comments are greatly appreciated. As suggested, the words were revised in the revision. (see lines 40)
4. Change “According to the different direction of nucleotide chain, it can be divided into” to “Depending on the 5’-3’ versus 3’-5’ orientation of segments in the self-folded oligonucleotide chain(s), G quadruplexes can be categorized as” in lines 41-42.
Reply: According to your meaningful suggestions, the sentence was revised in the revision. (see lines 41-42)
5. The addition of Figure 8 is meaningful, but this reviewer does suggest that the wording in the Figure caption should be revised to simply state a brief description of the data (e.g., Figure 8. Micrographs of nuclear staining of cell monolayer for control group (A)…), not a discussion of what can be inferred or concluded from the experiment (e.g., exclude the phrase “Detection of apoptosis”)
Reply: According to your meaningful suggestions, we have revised the figure caption. (See Figure 8 in line 364-365)
6. In the Introduction section, the background and significance of virtual screening are not described sufficiently. Please revise.Figure 9 is the only example in which the authors did not appear to make a suggested change from this reviewers previous comment. Since they also did not explain why, this reviewer is still of the opinion that the x-axis in Figure 9 (renumbered in revised manuscript) should be labeled as “Relative Fluorescence Intensity” rather than with a unitless “DNA content” label.
Reply: We greatly appreciate your valuable suggestion. Figure 9 and its caption were revised in the revision. (See Figure 9 , and it caption in lines 382-386)
Reviewer 2 Report
The authors have addressed carefully all the concerns from the reviewer. The overall quality of current manuscript is greatly improved, therefore it's recommended for publication in Pharmaceutics.
Author Response
Thanks a lot for your nice comments.